# Molecular Asymmetry in Prebiotic Chemistry: An Account from Meteorites

**DOI:** 10.3390/life6020018

**Published:** 2016-04-13

**Authors:** Sandra Pizzarello

**Affiliations:** School of Molecular Sciences, Arizona State University, Tempe, AZ 85287, USA; pizzar@asu.edu; Tel.: +1-480-965-3370

**Keywords:** meteorites, asteroids, abiotic, prebiotic, chemical evolution, enantiomeric excess

## Abstract

Carbonaceous Chondrite (CC) meteorites are fragments of asteroids, solar planetesimals that never became large enough to separate matter by their density, like terrestrial planets. CC contains various amounts of organic carbon and carry a record of chemical evolution as it came to be in the Solar System, at the time the Earth was formed and before the origins of life. We review this record as it pertains to the chiral asymmetry determined for several organic compounds in CC, which reaches a broad molecular distribution and enantiomeric excesses of up to 50%–60%. Because homochirality is an indispensable attribute of extant polymers and these meteoritic enantiomeric excesses are still, to date, the only case of chiral asymmetry in organic molecules measured outside the biosphere, the possibility of an exogenous delivery of primed prebiotic compounds to early Earth from meteorites is often proposed. Whether this exogenous delivery held a chiral advantage in molecular evolution remains an open question, as many others regarding the origins of life are.

## 1. Carbon-Containing Meteorites and Cosmochemical Evolution

One of the paradigms of chemical evolution proposes that the abiotic formation of increasingly complex molecules throughout cosmochemical and Earthly processes could eventually lead to life’s precursor molecules. It finds its rationale in the observation of phenomena, both leading to, and proceeding from, the origin of life, such as the formation of complex organic molecules in interstellar media (e.g., [1]), and a terrestrial phylogeny that evolved from much simpler organisms (e.g., [2]).

To date, such chemical evolution has found direct experimental scrutiny, solely through the analyses of Carbonaceous Chondrites (CC) meteorites, stony fragments of asteroids containing abundant organic carbon, ranging in complexity from kerogen-like macromolecular materials to simple soluble compounds, several of which are identical to terrestrial biomolecules. Because CC Asteroidal parent bodies are planetesimals found between Mars and Jupiter that never accreted large enough to undergo the extensive differentiation involved in planet formation, their meteoritic fragments uniquely offer a pristine record of abiotic organic chemistry in the early Solar System, as found in a planetary setting, at a unique juncture in the long history of the biogenic element, and at its closest to the onset of life. Their studies, therefore, have offered an understanding, not only of how compounds identical to life’s molecules were formed abiotically, but also about the organic inventory accreted by early Earth, e.g., through its heavy bombardment from comets and meteorites [3,4].

Comprehensive molecular and isotopic analyses for over forty-five years, and continuing today, have revealed that CCs may contain a complex organic suite of compounds, which are seen as the products of the synthetic capabilities of both solar and pre-solar environments [5]. Their complex range varies from polar compounds, such as amino acids, to non-polar hydrocarbons (e.g., [5,6,7]) and, when detected simply by their elemental compositions, may include thousands of molecular species [8]. From the perspective of a possible continuity between cosmochemical and prebiotic evolution, this diverse suite would not appear to indicate any selective synthetic trend and, in fact, indicate a fundamental distinction between abiotic and biological chemical processes in their gaining of molecular complexity. To say it with the words of Shakespeare, therefore, a meteoritic organic input of thousands H, C, N, O containing molecules would have offered “too much of a good thing” for an evolutionary transition towards the origins of life.

Nevertheless, within this evident heterogeneity, several meteoritic compounds have been found to contain L-enantiomeric excesses (*ee*), *i.e.*, to be more abundant in the l-chiral form like the amino acids of terrestrial proteins [9], offering the first unequivocal indication that a purely chemical evolutionary process, as the one believed to be responsible for the formation of meteorite organics, may also have been at work in chiral selection, a property known since the time of Pasteur to be intimately associated with life.

The following review reports on *ee* distributions found within meteorite organics so far, the possible venues that might have lead to their formation, and the effect that the input of such selected molecules might have had in prebiotic chemistry.

## 2. Enantiomeric Excesses in Meteoritic Compounds

The search for chiral asymmetry in meteorites was first undertaken by Engel and Nagy [10] and, although carried out carefully, was controversial [11] because of the pervasive distribution of homochirality in the biosphere and the possibility of terrestrial contamination in carbonaceous meteorites resulting from bacteria, molds, or their remnants. However, *ee* were ultimately detected in the Murchison meteorite for some amino acids not common in terrestrial proteins and abundant in meteorites, the 2-methyl, 2-aminoacids (2maa), and to have the same configuration (L) as protein amino acids [9]. To date, L-*ee* have been detected for a number of amino acids, as well as other compounds, and their inventory is summarized in Table 1 [12]. As shown, the larger number comprises the amino acids of several meteorites and involves molecular species that lack an α-H, with the exception of the diastereomers of isoleucine, as further detailed below, and the one-time finding for aspartic acid [13]. Lactic acid is the only hydroxy acid to display *ee* [14], while two recent communications have reported their detection in several sugar acids derivatives [15] and two amines, *sec*-butyl- and 1-methylbutyl amines [16]. These latter studies, appear to point at the possibility of diverse asymmetric effects in cosmochemistry. These meteoritic enantiomeric excesses are still, to date, the only case of molecular asymmetry measured outside the biosphere.

The overall extent of *ee* in 2maa varies significantly, between 0% and 18%, and even within short distances in the same meteorite stone [17,18]. For Isovaline (2-amino, 2-methylbutanoic acid), the most abundant of the acids, the absence of any contribution from terrestrial contaminants within this chiral heterogeneity was validated by Murchison, using ^13^C-, and D isotopic analyses [17,18,19], where the two enantiomers were found to have comparable enrichments. Isovaline δD values were particularly high, +3600‰ [19], and near to those measured spectroscopically for organic molecules in the interstellar medium.

It should be noted that racemization, the conversion of one enantiomer into another, is possible for 2H-amino acids in water through a repeat statistical abstraction and reacquisition of the α-H and because of its vicinity to the electron-withdrawing carboxyl group. Racemization, with time and proper conditions, will lead to racemic solutions (equal amounts of the two enantiomers). The fact that all amino acids found non-racemic in meteorites have a 2-methyl in place of a 2-H and cannot racemize, easily raises the question of whether the twelve racemizable meteoritic amino acids detected having terrestrial protein counterparts might have initially carried *ee* but racemized during the water phases known to have affected CC parent bodies. The understanding of the processes involved in meteoritic amino acids’ syntheses and the apparently odd *ee* findings for diastereomer amino acids in several meteorites may be able to address the question.

A possible pathway proposed for the formation of 2-amino-, and 2-hydroxy acids in meteorites is the addition of ammonia and HCN to ketones and aldehydes in the presence of water (Figure 1a, Reference [12] and the references therein). This is a reasonable hypothesis, for at least some of the amino acids because all the needed reagents are abundant components of interstellar ices and likely constituents of primitive asteroids [20,21]. Although producing an asymmetric carbon and, therefore, a chiral molecule in most cases, this type of synthesis is non-stereospecific because the HCN addition would be random and give equal amounts of D-, and L-enantiomers.

However, reaction products become more complex for longer aldehydes that already contain an asymmetric carbon, such as in the synthesis of the 6-C isoleucine (ile) and *allo*isolucine (*allo*) diastereomers (Figure 1b) from DL 2-methylbutanal [12]. In this case, were an *ee* already present in the precursor aldehyde, e.g., of the (S) configuration, those diastereomeric product amino acids that carried the (S) portion of the precursor through their synthesis will be more abundant than their respective enantiomers and any original *ee* would be revealed in different compounds.

In the above example, this would be the (RS) *allo* and (SS) ile compounds or, in the formalism used for amino acids, D-*allo* and L-ile. As can be seen, these two sets of diastereomers are also each other’s racemization product (called epimerization in this case,) and, because the 3C is too far removed from the carboxyl and will racemize much more slowly, if at all, original *ee* may be preserved much longer.

Such was the distribution found in the extracts of a group of pristine meteorites collected in Antarctica (Renazzo-type or CR) (Figure 1c, [22,23]) and, based on the above formative premise, as well as upon confirming the indigeneity of all four individual enantiomers by ^13^C isotopic analyses [22], the findings were interpreted to signify that their precursor aldehyde carried an original S-asymmetry to the meteorite’s parent body. CR meteorites’ organic composition, therefore, offers a new exciting account of cosmochemical evolution’s capabilities toward the selective production of very large abiotic *ee*, of up to 60% in the case of the isoleucine diastereomers, in some meteorites (Table 1).

## 3. The Likely Locals for Asymmetric Syntheses

The origin of *ee* in the amino acids of meteorites has been long debated since their initial detection, and is still not known (e.g., [24]). The first hypotheses to be put forward proposed their possible asymmetric decomposition upon UV photolysis [25] and, following the hint of amino acids’ high isotopic enrichments in deuterium, the likelihood of their syntheses in cold, deuterium-enriching cosmic environments [7]. The two enantiomers of a chiral molecule have identical chemical properties, but differ in some physico-chemical features, e.g., polarizability [26], and, as a result, may react/interact differently with other chiral molecules or entities (a common illustration could be that of the distinct ways two people can join hands for different aims).

The effects of UV circularly polarized light (CPL) irradiation, which can have right-handed or left-handed helical paths of propagation (Figure 2), were discussed and studied in detail [27] and for amino acids in particular. Several experiments did show UVCPL irradiation generating adsorption bands by the compounds with the same sign as the irradiating CPL [28], and even over a large wavelength range [29], however, the maximum *ee* achieved were found to be, at most, 9% (at an unlikely pH 1 [30]), *i.e.*, far lower than found in meteorites.

These experimental data leave open the possibility that *ee* were first formed in precursor molecules, such as the aldehydes mentioned above, which would also justify the findings for ile and *allo*. Aldehydes have not been studied in this regard, e.g., their circular dichroism is not known, nor is their behavior upon CPL irradiation, and the field is, in fact, yet open to inquiry. However, the proposal of their participation as precursors to the syntheses of amino acids in meteorites would draw the important inference that, because chiral aldehydes racemize quickly in water, in fact, much more rapidly than amino acids, the synthetic window for amino acids in meteorites from non-racemic aldehydes during the asteroidal water processes should have been short and icy [22]. Early parent body aggregates or protoplanetary disc environments would fit the scenario, with ammonia abundance possibly providing a lower freezing point of water.

Although the scenario also implies that meteoritic *ee* would be rare and could be detected for 2-amino acids only in the special cases, such as for the diastereomer compounds described above, the hypothesis does not explain the *ee* for 2maa, of which precursors would be ketones that *cannot* racemize. Additionally, the large heterogeneity of *ee* values demonstrated throughout these studies of meteoritic chiral compounds is very puzzling and has brought other proposals that the formation of *ee* could be related to asteroidal parent body effects during aqueous stages [12,31] and, in particular, catalysis from their mineral phases.

This was first addressed to justify the large variability of isovaline *ee* between contiguous fragments of the Murchison meteorite and, by their X-ray diffraction analyses, showed a possible correlation between isovaline *ee* and hydrous silicates abundances [16]. In the case of CR meteorites mentioned above, that their aqueous processes are found to be variable within the meteorite matrix [31] would also satisfy the observation of varying *ee* detected between meteorites’ samples.

The recent findings of two non-racemic amines in several CR2 meteorites with *ee* as high as 60% [15] were able to further this discussion. Because, as for amino acids, amines’ circular dichroism [15] and anisotropy spectra [32] do not encourage suggestions of possible asymmetric effects by UV CPL, it has been proposed that the ketones precursor to these amines, while not chiral, could have benefited by adsorption upon mineral phases possessing asymmetry. Such minerals, e.g., magnetite seen to have in meteorites helical structures [33], could then provide chiral induction during syntheses. For 2-methylamines, it could have been a reductive amination process made possible by the large abundance of ammonia known to be present in this type of meteorites, e.g., [34]. If further confirmed experimentally, processes like these would point to possible important roles for Asteroidal bodies, their aqueous history and the minerals ensuing from warm hydrous stages in the development of chiral asymmetry in prebiotic molecules.

## 4. Prebiotic Chiral Asymmetry and the Origins of Life

The origins of life are utterly unknown, have been debated for close to a century, and the only achieved consensus is one of life likely emerging from inanimate molecules, about 3.5 billion years ago and by diverse possible scenarios, such as syntheses in the early atmosphere, the deep Earth or by input of organics materials from comets and meteorites (e.g., [5,35,36]). As mentioned already, and as Cronin and Chang [37] remarked pointedly, the known meteoritic inventory discussed above “…is clearly not the recipe for constructing the progenote…and a basic challenge resides in finding…where diversity that characterizes chemical evolution…gives way to the selectivity of life”. The question therefore is: Has the molecular asymmetry of meteoritic compounds changed the odds of an exobiology?

One distinguishing characteristic of Earth life is that the structure and functions of biopolymers rely on the exclusive one-handedness of their monomeric components, *i.e.*, most protein amino acids have an L-configuration while sugars in RNA and DNA have a D-configuration and substitutions along the polymers with enantiomers of opposite handedness usually result in loss of function. That the trait of molecular asymmetry could be, or become by evolution, so pervasive as to define life processes and that several meteoritic compounds may have *ee* of the same L-configuration as terrestrial proteins do seem to provide a new dimension to this debate.

However, the overall data also pose a caveat. As we have seen, the asteroidal aqueous environments, as recorded in the mineralogy of all meteorites containing organic compounds of prebiotic interest, tend to racemize either the α-H amino acids or their aldehyde precursors, making *ee* for these amino acids scarce in the mature parent bodies. The α-substituted species that do not racemize would be more abundant but possibly less reactive. On the other hand, their ketone precursors do not racemize at all and could, in fact, have been very desirable prebiotic reactants. Moreover, some of the CC meteorites, such as the Renazzo-type, contain, not only abundant amino acids, but also large abundance of ammonia [38,39,40], an indispensable reactant for their syntheses. The combination of mineral catalysts with exogenously delivered key ingredients and chiral compounds, therefore, may still represent a very good evolutionary chance in prebiotic chemistry.

## Figures and Tables

**Figure 1 life-06-00018-f001:**
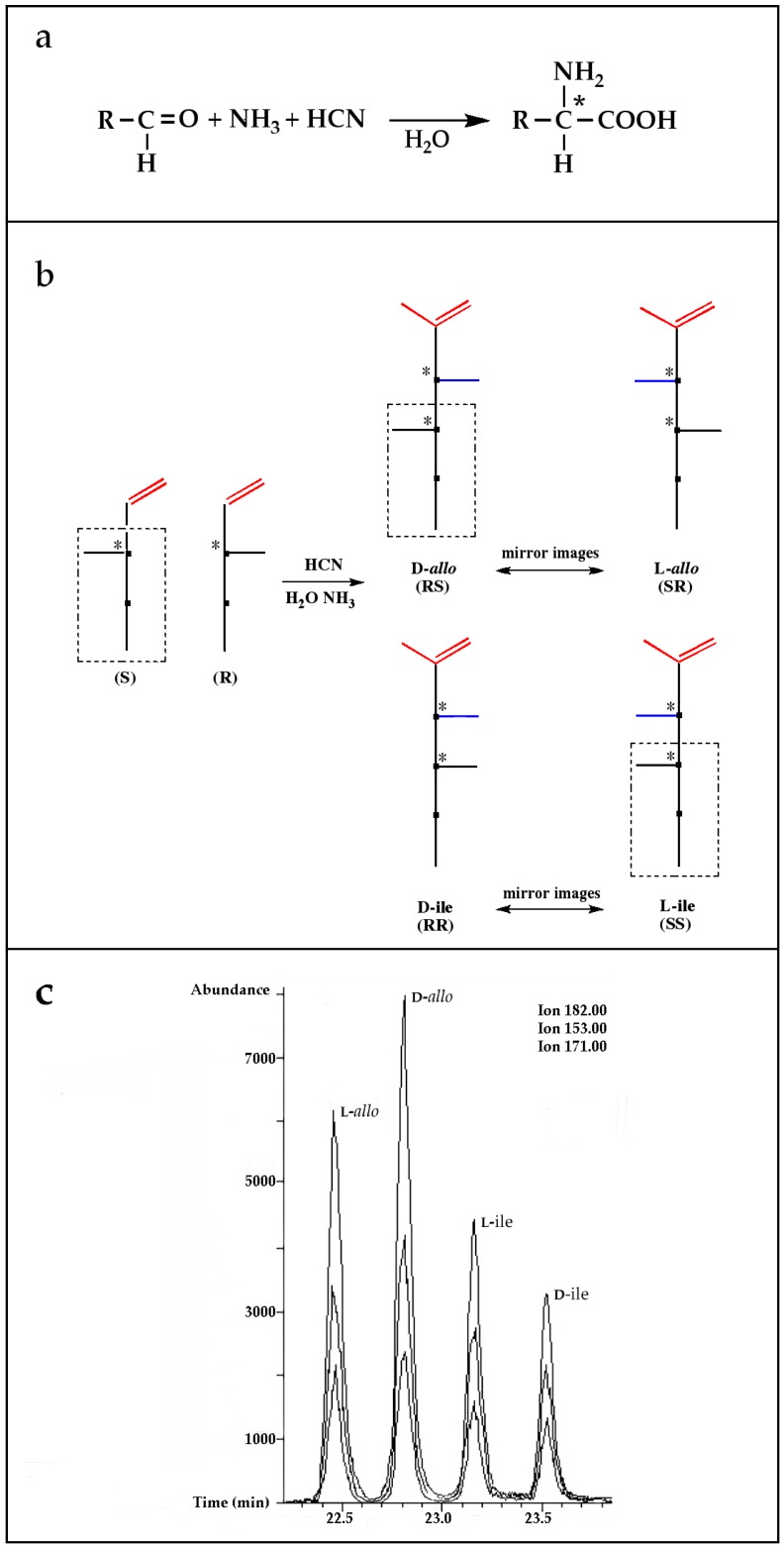
(**a**) The Strecker synthesis for the possible formation of amino acids in meteorites ([12] and references therein); (**b**) Structural relation of the *allo*isoleucine and isoleucine diastereomers; (**c**) Enantiomeric excesses detected for the isoleucine diastereomers in the GRA 95229 meteorites using Gas Chromatography-Mass Spectrometry, single ion traces [22].

**Figure 2 life-06-00018-f002:**
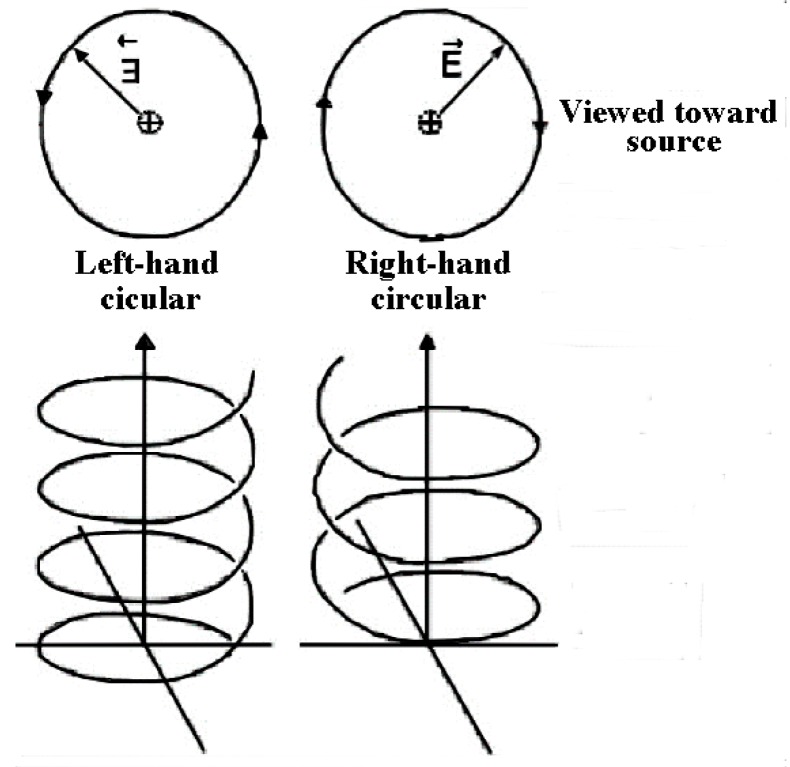
Graphic representation of Left and Right Circularly Polarized Light.

**Table 1 life-06-00018-t001:** Amines and amino-, hydroxy, and sugar-acids displaying enantiomeric excesses in carbonaceous chondrites.

Compound Formula	Compound	*ee*% ^1^	Distribution ^2^
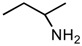	*Sec*-butylamine	(S) 0-18	GRA 95229 ^3^^,^* LAP 02342 MIL 07525 PCA 91082 EET 92049
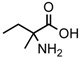	2amino2methylbutanoic a. ^4^ (isovaline)	L 2.5-19.6	MN, MY *, Or ^5^ LAP02342 ^6,^*
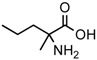	2amino2methylpentanoic (2methylnorvaline)	L 1.4-2.8	MN, MY
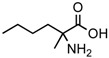	2amino2methylhexanoic (2methylnorleucine)	L 1.8	MN, MY
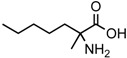	2amino2methylheptanoic	L	MN
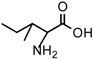	2amino3methylpentanoic (isoleucine)	L 4-50	MN, MY GRA 95229 LAP 02342
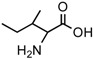	2amino3methylpentanoic (*allo*isoleucine)	D 2-60	MN, MY GRA 95229 LAP 02342
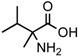	2amino2,3methylbutanoic (2methylvaline)	L 1.0	MN, MY
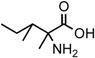	2amino2,3methylpentanoic	L 1.4-5.2	MN, MY
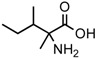	*allo* 2amino2,3methylpentanoic	L 2.2-10.4	MN, MY
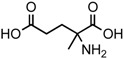	2methylglutamic	L 3-2	MN
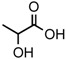	Lactic	L 3.0-12-3	MN ^7,^*, GRA95229 * LAP 02342 *
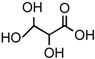	Threonic	D	MN ^8^

^1^ Values from [12] unless otherwise indicated; ^2^ Meteorites where *ee* of the given compounds are found. MN, Murchison; MY, Murray; Or, Orgueil; * isotopic data acquired for individual enantiomers; ^3^ [16]; ^4^ acid; ^5^ [18]; ^6^ [39]; ^7^ [14]; ^8^ [15], this last study detected other *ee*-containing sugar acids.

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
