# Peer review of "Molecular Asymmetry in Prebiotic Chemistry: An Account from Meteorites"

_life, 2016, doi:10.3390/life6020018_

Reviewer 1 Report

Topical and well-written article that reviews the chirality of meteoritic amino acids and their chemical formation. 

In line 82, it might have to read "L- and D-isovaline" instead of "L- and DL isovaline". Please check. In Figure 1c the unit of the x-axis is missing. To address readers from related disciplines the author might cite in line 128 the book "Meierhenrich: Amino Acids and the Asymmetry of Life, Springer 2008". The paragraph on aldehydes (lines 145-152) should cite new data on the detection of aldehydes in simulated interstellar ices "de Marcellus et al.: Proc. Natl. Acad. Sci. USA 112 (2015), 965-970" and also identified by the cometary Rosetta-Philae mission "Goesmann et al.: Science 349 (2015), 497, aab0689". 

Author Response

Thank you for the kind comments. Yes, the isotopic comparison was between L-, and DL isovaline and I see the axis on the figure. All suggested new references were added

Reviewer 2 Report

The topic of the review article, insights into molecular asymmetry from meteorite amino acids, is definitely interesting and important for the diverse readership interested in origins of life in Life journal. However, the manuscript primarily discusses the author’s own work, and essentially ignores all other relevant work performed by other groups in the last two decades. In fact, there is only one reference dated between 2011 and 2016 (Schrader et al. 2015) that does not list Pizzarello as an author. The exclusion of the amino acid work by the groups of Meierhenrich (e.g., Myrgorodska et al. 2016 Journal of Chromatography A), Martins (e.g., Martins et al. 2007 Meteoritics & Planetary Science), Dworkin (e.g, Glavin et al. 2010 and 2012, Meteoritics & Planetary Science; they have apparently published on this topic extensively since 2006 or so) and Herd (e.g, Simkus et al 2016 LPSC abstract #2370 on aldehydes and ketones), as well as interesting work to explain the origins of chirality by MacDermott and others (e.g., MacDermott et al 2009 OLEB; MacDermott 2012 Comprehensive Chirality 8, pp 11 – 38). The omission of all of these works, particularly those published after 2011, would be a detriment to the readership of Life. Thus, the current manuscript as presently written, is merely an incremental update of previous review articles written by the author (e.g., Pizzarello and Shock 2010, Pizzarello and Groy 2011), and does not merit publication in Life.

To be considered for publication, the manuscript should be adapted to include the relevant recent work by other authors, as well as addressing the points below.

-Table 1. Several minor changes to the table should be made to enhance readiblity. The use of ‘a.’ does not save much space, but requires the reader to read through the footnotes to determine that this is short for ‘acid’. The amino acid names should follow IUPAC convention, and use dashes to separate words and numbers ‘2-amino-2,3-dimethylbutanoic acid’. The way the ee% values are reported is unfriendly. Is the ‘-‘ after the L meant as a spacer or is it a minus sign for the first value in ranges? Can an emdash ‘–‘ or the word “to” be used to denote the ranges? If the emdash is used, can ‘L’ be placed in parentheses after the values. So for isovaline, ‘2.5 – 19.6 (L)’ rather than ‘L-2.5-19.6’. Also, for 2-methylglutamic acid, is the ee range -3 to +2? If not, why is the number larger in magnitude listed first? Why is there no value of ee for threonic acid or 2-amino-2-methylheptanoic acid? Rather than use arbitrary abbreviations for the non-Antarctic meteorites (MN, MY), why not use the full names, particularly since the readership of Life journal is likely not to be meteorite experts.

Page 3 – in regards to the +3600 delta D values, it is unclear what the significance of having “the highest ever recorded for organic molecules besides those measured spectroscopically for interstellar compounds.” This clause is also inconsistent with delta D values reported in Pizzarello and Holmes 2009 (2-aminoisobuytric acid and isovaline) and Elsila et al 2012 (alpha-aminoisobutyric acid, isovaline, etc.). The isotope discussion is also missing context for non-specialists, such as ranges for terrestrial isotope ratios.

Page 3 – “The fact that all amino acids found non-racemic in meteorites have a 2-methyl in place of a 2-H and cannot racemize…” is contradicted by the isoleucine and allo-isoleucine entries in table 1, which do not contain alpha methyl groups. There was a report by the Dworkin group about indigenous enantiomeric excesses of L-aspartic acid as well, which does not have an alpha methyl group.

Figure 1 is identical with figure 5 from Pizzarello and Shock 2010, and nearly identical to Figure 2 from Pizzarello and Groy 2011, with the exception that that the hetero atoms are labeled in the version in Pizzarello and Groy. Of the two previously published figures, I prefer the one with the heteroatoms labeled because it otherwise implies that the molecules are only color-coded, chiral alkenes. The figure legend should state where the figure was originally published, whichever one is chosen. Also, the word “non-superimposable” should be added above “mirror images” to indicate that they are enantiomers and not identical molecules, since this is for a general audience and not exclusively chemists.

The manuscript presents isoleucine as an idealized case for nature producing very large enantiomeric excesses. However, because you are starting with a chiral precursor, there will be a free energy difference between the two possible ways to add ammonia in the Strecker-cyanohydrin synthesis. A Strecker-based route from a chiral precursor only applies to a small fraction of the compounds found in enantiomeric excess listed in Table 1. There a three possibilities, then: the route from chiral aldehyde to isoleucine molecular asymmetry is the only way to produce molecular asymmetry and isoleucine initiated homochirality on Earth; there are multiple mechanisms to molecular asymmetry and isoleucine followed a rare one not available to most amino acids formation pathways; or the proposed route beginning with aldehyde asymmetry was not followed, and molecular asymmetry was initiated through a mechanism available to all amino acid precursors (the option favored by Occam’s razor). The limitations of the isoleucine analyses for explaining the observed chiral excesses in a wide range of molecules including those presented in table 1 should be explicitly stated in the manuscript.  

While isoleucine offers analytical advantages over other proteinogenic amino acids in that it epimerizes to a diastereomer rather than its enantiomer, the isoleucine stereoisomer distribution revealed by meteorite analysis has little biological utility. The most important aspect of homochirality for proteins and peptides is the chirality at the alpha-carbon, as this is makes the protein backbone and guides the allowable phi- and psi- angles that dictate protein folding into alpha helices, beta sheets etc. Thus the more important comparison for isoleucine stereoisomers is the ratio of D-alloile to L-Ile; in all cases the D-alloile stereoisomer is equally or more abundant than the L-ile stereoisomer (e.g., Pizzarello et al. 2012 PNAS). If the first peptides included a stereoisomer of isoleucine, it would be more likely to be D-alloisoleucine than L-isoleucine, just due to the higher presence of D-alloisoleucine. Considering that all other molecular asymettries previous reported are L-excesses, this suggests that the set of isoleucine diastereomers were minor players in the origins of homochirality, or they worked actively against it. The implications of the predominance of D-alloisoleucine over L-isoleucine should be discussed in the manuscript.

Page 5 – the Meierhenrich  and de Marcellus group presented at a conference recently that  aldehydes, ketones and amines do not show anisotropy / circular dichroism differences under UV-CPL irradiation. The author may wish to contact them to get more details, but it appears this hypothesis is no longer valid. 

Author Response

Version:1.0 StartHTML:0000000182 EndHTML:0000009427 StartFragment:0000002598 EndFragment:0000009391 SourceURL:file://localhost/Users/pizzar/Dropbox/2016/Luisi/Revs_respons.doc

This is an experimental review and, although referring briefly to the possibility of asymmetry effects in interstellar environments, such as decomposition by UV CPL irradiation, I did not consider appropriate going into theoretical debates and afield from my expertise. I know Alexandra MacDermot well and personally but she would better write her own review of theoretical work, hers and of others. I changed the title of this review accordingly.

The work of Meierhenrich has been quoted with two added references plus that to his book and I certainly would not want to enter the topic of analytical methodologies with Myrgorodska et al., 2016. This paper is interesting for conoscenti maybe but not ground shuttering and, before that, a whole article would be needed to report all methods currently or previously utilized, that would assuredly be detrimental to this review article…..

Of the references suggested by this reviewer, Martins et al. 2007 did not report enantiomeric excesses, in fact, the authors point to the fact that D, L isovaline in the CR analyzed could not be separated chromatographically (I will agree, methodologies matter). The 2010 work by Dworkin and Glavin was largely a report of amino acid abundances, the data for ee are burdened by very large errors and I did not think it useful as a reference.

As for referencing the 2012 paper by Glavin et al. I referenced it but under duress. It is published, in a geochemical journal, and has been questioned because in a previous manuscript, just one year earlier (Herd et al., 2011), the authors analyzed “the same samples” (their statement) without reporting any excesses for L-aspartic, albeit reporting ee for other compounds, and much can happen to a sample in a year in the way of contamination. Obviously, if the samples had displayed large L-aspartic ee, it would have been reported in Science?

However, I referenced freely the same authors’ 2011 paper reporting detection of isovaline ee in the Orgueil meteorite, it also proposed the wrong explanation for their attainment, which prompted me to clarify it with my 2011 paper with Groy.

Table 1. I apologize for the forgotten numbers, they have now been added. Chemical names were expanded to IUPAC conv. As for the shortening of long and obvious names, my first thought reading this reviewer’s comment was that any reader puzzled by the meaning of a. next to an glaring acid’s structure would be better off skipping the table all together. But then, gathering from the general tone by the reviewer, I concluded to be unlikely to convince.

I changed the title of the table to take the incriminating “a.” out. MN and MY are common acronyms for the two meteorites, and yes they are there for a reason, i.e., the length of their full names, plus these are explained in the legend. Furthermore, as the reviewer acknowledges, acronyms are often used even officially to shorten long names, e.g., GRA stands for Graves Nunataks, LAP for La Paz Icefield, etc.

As for pg. 3 first question, this is not really the place to expand for the reviewer either on the significance of referenced D/H enrichments detected in meteoritic compounds or about the occurrence of heterogeneity in CC meteorites’ organic and mineral compositions, suffice to say that the first finding was considered quite significant by the community and the latter is an experimental matter of fact. All is referenced.

For 2-methylaminoacids, the ile/allo exception in CC is clearly stated following the statement highlighted by rev3 and so is now the aspartic new reference entered against my best judgment.

There is a request pending for permission by CSH for Fig.1, when I hear back, I will either forward the permission to Life’s office or modify the Figure.

The comments of the reviewer on ile/allo, the compounds’ biological utilization, the importance of their relative abundance etc. are interesting, as stated already however, this is an experimental report, what is reported are findings in meteorites and, for these amino acids in particular, how they fit the discourse on the distribution of ee in prebiotic chemistry. The simplest and reasonable conclusion, which the reviewer finds simplistic, is that asymmetric effects could have affected 2-H amino acids and that those effects created large ee. The reviewer is quite welcomed to pick up from there in a manuscript of his/her own.

I recently measured (ref.15) the CD of sec-butylamine and its absorption would leave open its asymmetric decomposition but, thank you for the suggestion, I will gladly discuss it with dr. Meirhenrich at the first occasion.

Reviewer 3 Report

The review agrees with the objectives of the journal. The author is highly representative in topic. The discussion is sound and the text good write and it presents a personal and reliable point of view discussing the aspects concerning racemization vs structure and origin of life.

In summary I strongly recommend the text for publication. 

Minor points

1) In the discussion of section 4. in the note on the input of organic materials from comets and meteorites to Earth I miss the citation of the old original reports in addition to ref. [27]

2) I do not see any philological reason to write “a-biotic, a-biotically, etc” instead of the historical and gramatically correct term “abiotic”

Author Response

I did not quite know what reference the reviewer wanted added, for meteorites, that would have been Kvenvolden et al. 1970, however the work was alredy in the eviews quoted (ref.s 5-7), therefore I referenced Sandford et al. 2006 for comets. I would be happy to revise again.

Round  2

Reviewer 2 Report

The manuscript has been improved but it still contains many significant issues that need to be corrected prior to its publication, including several issues raised in the first review that were not adequately addressed. The manuscript also still represents only a minor update over previous reviews by the author, including Pizzarello and Groy 2011 and Pizzarello and Shock 2010, as it draws largely from a single paper published since those two, the Pizzarello et al. 2012 PNAS paper.

-If the errors of ee values from Glavin et al 2010 are too large for them to be useful, then papers that do not address errors in ee data at all or even discuss whether or not replicate measurements were made should also be removed from this review. Therefore, Pizzarello et al 2012 PNAS, which does not discuss errors for amino acid abundances or enantiomeric excesses, nor does it address replicate measurements that would permit errors to be measured. Pizzarello et al. 2010 should also be removed, because although, repeat injections of selected standards it cannot be determined that repeat injections of the actual samples being measured were performed.

-Table 1 is improved. However, it is still written for the convenience of the author rather than the sake of the reader. The chemical structures listed in Table 1 are far too small. Also, since sec-butylamine is not an acid, it should not have any entry in the “acid formula” column. The IUPAC names are missing the required ‘-‘ between letters and numbers. Values are still missing for 2-amino-2-methylheptanoic acid and threonic acid. The full names of Murchison, Murray and Orgueil should still be included, and ‘MN’, ‘MY’ and ‘Or’ are not official abbreviations. The difference between the abbreviations used here for those three meteorites and the Antarctic ones is that the Antarctic meteorite abbreviations were officially approved by the organization in charge of meteorite nomenclature, the Meteoritical Society. Also, when the Antarctic meteorite abbreviations are used, they must be spelled out in full in the text before the abbreviations are used, so these need to be spelled out fully in the text as well. The values listed for lactic acid are confusing: is the ‘-‘ notation used above to indicate a range now being used to separate items in a list: “3.0-12-3”? If so, the dash should be replaced by commas (“3.0, 12, 3”). If this is the case, why are individual values given for lactic acid but not for sec-butylamine or any of the other compounds where more than 2 meteorites were measured. Also, what does the “3-2” for 2-methylglutamic acid mean in Murchison? Does this indicate +3 to +2, or -3 to +2, or that separate measurements of this compound in Murchison gave values of +3 and +2? Again, the use of endashes (which actually mean subtraction or negative sign) to indicate range rather than emdashes or the word “to” makes the data presented in the table even more confusing.

-As the author points out in the response to reviewer, the reported delta D value of +3600 is an experimental matter of fact, but it is also an experimental matter of fact that Pizzarello and Holmes in 2009 reported delta D values of +7245 for 2-aminoisobutryic acid and +5807 (or +7050 if corrected for a co-eluting impurity). It is also an experimental fact that Elsila et al 2012 MAPS reported delta D values of +4303 and +7065 for alpha-aminoisobutyric acid and +6017 and +4956 for isovaline. It is also a fact that each of these values is larger than +3600. If the largest absolute magnitude of deuterium enrichment is sufficiently important to mention in this manuscript, the correct values should be used. Depending on whether or not one corrects for the co-eluting impurity, this would be +7245 for 2-aminoisobutyric acid and +6017 or +7050 for isovaline. Regardless of which highest-isotopic enrichment value is settled on as the correct one, the Elsila et al. 2012 MAPS paper, which reports isotopic ratios of amino acids in a wide breadth of meteorite types and amino acids, adds important data to the discussion of stable isotopic measurements and should be reference in this section as well. This whole section is still missing the two or three necessary sentences to indicate to someone not intimately familiar with heavy isotope enrichments of molecules formed in cold interstellar environments how the 13C and deuterium measurements rule out significant terrestrial contamination.

Page 4 – The statement “The fact that all amino acids found non-racemic in meteorites have a 2-methyl in place of a 2-H and cannot racemize” seems to contradict a fundamental premise of the paper, which is that isoleucine enantiomeric and diastereomeric excesses tell us something about the origins of enantiomeric excesses prior to the start of life. The wording should be changed, perhaps from “all” to “most”.

-Figure 1 still needs to include heteroatom labels. It should not be left to the reader to deduce that the red lines indicate a carboxylic acid, rather than an alkene, and that the blue lines indicate amines. It should also be indicated that the asterisks indicate chiral centers – the readership of Life journal includes scientists that are not chemists. The significance of the ions 182, 153, and 171 should be explained briefly in the legend. The decimal points should be removed as they imply precision not attainable with GC-quadrupole mass spectrometry, which has only unit m/z resolution.

-The anisotropy spectra for sec-butylamine by Meinert et al have now been published: “Anisotropy spectroscopy of chiral alcohols, amines, and monocarboxylic acids: implications for the analysis of extraterrestrial samples”, in Journal of Photochemistry and Photobiology A: Chemistry. This manuscript indicates that with 99.99% destruction of sec-butylamine would yield only a 0.18% ee, providing compelling evidence that UV-CPL is not a plausible route for the formation of enantiomeric excesses in amines. As the Pizzarello and Yarnes paper is not yet available, either the new Meinert et al. work should be referenced and included in the discussion, or the Pizzarello and Yarnes reference should be removed.

Author Response

I was (also) very pleased to add to Life review manuscript data on the first time finding of enantiomeric excesses for chiral amine in several meteorites. The data were referenced to a new manuscript of mine that was recently accepted and is currently in press (as your office had previously confirmed to be allowed; Pizzarello and Yarnes, 2016). This manuscript also offers a novel proposal for the attainment of enantiomeric excesses through planetesimal processes instead than Interstellar or nebular ones, such as irradiation by UV CPL long suggested in the literature. I see now that rev. 2 is far from pleased with my revision and has chosen to engage in a back and forth debate.

For example, the reviewer had previously found "unclear what the significance" of mentioning the deuterium enrichment of meteoritic isovaline was (missing the overall context of excluding terrestrial contamination) and I replied, maybe too curtly, that it was what it was. Could I have instead explained with a short addition what the isotopic values meant for the discussion of that particular reference? Yes, I could have but also the reviewer chose to fall into an editorial fit and seems now to have discovered many references for me to quote, not all pertinent, and is asking for a full debate on the subject....

There are several other paragraphs of new requests by this reviewer, all equally rumbling, but I will go directly to the final paragraph of the review, which was most significant for my decision.

The reviewer states:

"As the Pizzarello and Yarnes paper is not yet available, either the new Meinert et al. work should be referenced and included in the discussion, or the Pizzarello and Yarnes reference should be removed".

Two comments:

1, The Meinert et al. paper, a laboratory study on the circular dichroism of several organic compounds, could certainly be referenced in the submitted manuscript but does not contradict the discussion in Pizzarello and Yarnes, 2016, in fact it re-enforces the argument there as well as several
papers of mine and by others, going back at least to the Nishino et al., 2009 (referenced in the submission), that UV CPL cannot be considered the sole causing agent of enantiomeric excesses for most amino acids or amines in meteorites.

2. The reviewer's dialectic went also totally askew with the above statement, what would this reviewer do differently if he/she could read the "in press" manuscript, judge the validity of a work reviewed and accepted by a reputable Journal? Or does the reviewer think that I would misrepresent my own data?

Given the clearly attitude of this reviewer, the fact that to further engage him/her in a discourse appears unlikely to be fruitful and I do not intend to do so, I respectfully propose to the editor the option of my submitting a revise manuscript including only: 1_ a few words to give further context to the debated D-enrichment values (line 105) and 2_ a reference for Meinert et al. where pertinent (e.g., 3. The likely locals for asymmetric syntheses).